# Co-Harboring of Beta-Lactamases and *mcr-1* Genes in *Escherichia coli* and *Klebsiella pneumoniae* from Healthy Carriers and Backyard Animals in Rural Communities in Ecuador

**DOI:** 10.3390/antibiotics12050856

**Published:** 2023-05-05

**Authors:** Carlos Bastidas-Caldes, Emily Cisneros-Vásquez, Antonella Zambrano, Andrea Mosquera-Maza, William Calero-Cáceres, Joaquín Rey, Yoshimasa Yamamoto, Mayumi Yamamoto, Manuel Calvopiña, Jacobus H. de Waard

**Affiliations:** 1One Health Research Group, Facultad de Ingeniería y Ciencias Aplicadas, Biotecnología, Universidad de las Américas, Quito 170124, Ecuador; emily.cisneros.vas01@udla.edu.ec (E.C.-V.); maria.zambrano.burgos@udla.edu.ec (A.Z.); 2Ministerio de Salud Pública del Ecuador, Pastaza 170505, Ecuador; dranataliamosqueram24@gmail.com; 3UTA RAM One Health, Department of Food and Biotechnology Science and Engineering, Universidad Técnica de Ambato, Ambato 180103, Ecuador; wr.calero@uta.edu.ec; 4Unidad de Patología Infecciosa y Epidemiología, Facultad de Veterinaria, Universidad de Extremadura, 10003 Cáceres, Spain; jmrey@unex.es; 5The United Graduate School of Drug Discovery and Medical Information Sciences, Gifu University, Gifu 501-1193, Japan; yyamamot@gifu-u.ac.jp; 6Health Administration Center, Gifu University, Gifu 501-1193, Japan; myamamotot@gifu-u.ac.jp; 7One Health Research Group, Facultad de Ciencias de la Salud, Universidad de las Américas, Quito 170124, Ecuador; manuel.calvopina@udla.edu.ec (M.C.); jacobus.dewaard@udla.edu.ec (J.H.d.W.)

**Keywords:** *E. coli*, *K. pneumoniae*, colistin, *mcr-1* gene, multidrug resistance (MDR), extended-spectrum beta-lactamase (ESBL), carbapenemase, strain type (ST), Ecuador

## Abstract

Few studies have addressed drug resistance of Enterobacterales in rural communities in developing countries. This study aimed to determine the coexistence of extended-spectrum β-lactamase (ESBL) and carbapenemase genes in *Escherichia coli* and *Klebsiella pneumoniae* strains carrying the *mcr-1* gene in rural communities in Ecuador from healthy humans and their backyard animals. Sixty-two strains, thirty *E. coli* and thirty-two *K. pneumoniae* strains carrying the *mcr-1* gene were selected from a previous study. PCR were performed for the presence of ESBLs and carbapenemase genes. The strains were further characterized, and the genetic relationship was studied with multi-locus sequencing typing (MLST) of seven housekeeping genes. Fifty-nine of the sixty-two *mcr-1* isolates (95%) harbored at least on β-lactam resistance gene. The most prevalent ESBL genes were the *bla*_TEM_ genes (present in in 80% of the *E. coli* strains) and the *bla*_SHV_ gene (present in 84% of the *K. pneumoniae* strains). MSLT analysis revealed 28 different sequence types (ST); 15 for *E. coli* and 12 for *K. pneumoniae*, with most ST never described in humans and animals. The coexistence of *mcr-1* and β-lactams resistant genes in *E. coli* and *K. pneumoniae* strains is alarming and threatens the efficacy of last-resort antibiotics. Our findings highlight backyard animals as a reservoir of *mcr-1/*β-lactams resistant genes.

## 1. Introduction

The emergence of multidrug-resistant (MDR) Enterobacterales isolated from humans and animals has become a great concern worldwide, and resistance to colistin in coexistence with β-lactams resistant genes compromises the effectiveness of antimicrobial drugs. Multidrug-resistant (MDR) Enterobacterales such as *Escherichia coli* and, more recently, *Klebsiella pneumoniae* have emerged as one of the significant causes of healthcare-associated communitywide infections [1,2]. It has also been shown that these MDR bacteria not only have shown up in hospital settings but can be isolated from healthy humans or from backyard animals, and both hosts can act as a reservoir for resistance genes [3,4]. 

During the last few decades, the emergence of colistin-resistant isolates has been frequently reported. The antibiotic colistin is a highly effective drug against most Enterobacterales species and non-fermenting Gram-negative bacteria and is considered a last resort antimicrobial for the treatment of difficult-to-treat infections caused by MDR. However, the uncontrolled use of colistin in human and veterinary medicine has resulted in the emergence of colistin resistance. Of serious concern are plasmid-borne or mobile colistin resistance (*mcr*) genes, whose first variant (*mcr-1*) was reported in China in 2016 [5]. These *mcr* genes assist in the dissemination of colistin resistance to other pathogenic bacteria. After this first report, the *mcr* genes have been found in more than 70 countries and 10 different variants have been described (*mcr-1*–*mcr-10*). In Ecuador, several recent studies have shown an increase of the prevalence of *mcr* genes in different hosts and sources [6,7]. The widespread of these resistance genes can be attributed to selective pressure due to the use of colistin in livestock and, consequently, the increase of the horizontal transfer of plasmids that contains this type of resistance gene [8].

Another worrying phenomena is that the plasmid-mediated resistance to colistin can co-occur with other resistance genes such as extended-spectrum beta-lactamases (ESBL) and carbapenemases [9,10]. The appearance of this type of resistance is also largely due to the irrational use of antibiotics for preventive prophylactic and/or growth-promoting purposes [11,12]; and threatens the One Health approach that seeks the well-being of humans and animals [13]. Beta-lactam antibiotics are used worldwide to treat Gram-negative bacterial infections and represent about 65% of the total antibiotics on the market [14]. The extensive use of those antibiotics for medical, veterinary, and animal-production purposes has triggered a natural selection of ESBL- and carbapenemase-gene–producing microorganisms as pathogens. These phenomena have caused resistances against cephalosporins, penicillin, carbapenems, and monobactams. ESBL genes are ubiquitous, with ESBL-producing Enterobacterales having been found in a wide range of settings—including livestock [15], companion animals [16], hospitals (involving human-to-human transmission) [17], the environment [18], and food—as well as being spread through travel [19]. Although β-lactamases are widely detected and reported in hospital areas, those same enzymes are not so frequently reported in animals, but mainly in those intended for food production [20].

CTX-M-type enzymes have surpassed TEM and SHV as the most prevalent group of extended-spectrum β-lactamases (ESBLs) and have been reported in various members of the Enterobacterales order, as well as in *P. aeruginosa* and *Acinetobacter* species [21]. In South-American countries, CTX enzymes are also more common than other ESBL types [22,23]. These β-lactamase CTX-M-producing bacteria have emerged globally as the primary cause of urinary tract infections, leading to what is now known as the “CTX-M pandemic” [24]. Among the CTX-M variants, the CTX-M-15 type within the CTX-M-1 sublineage is particularly concerning due to its heightened activity against ceftazidime and its association with other β-lactamases, such as VIM, OXA, and KPC [25]. Additionally, several dangerous clones have been identified, including the *E. coli* clones ST131 and ST10 [26,27].

The presence of both mobile colistin-resistant (*mcr*) genes and β-lactamase-encoding genes in Enterobacterales presents a significant threat to global health, as their dissemination could lead to untreatable infections, increased mortality rates, prolonged hospital stays, and higher healthcare costs. Therefore, monitoring the spread of these resistance genes is critical. Despite the importance of this issue, the co-occurrence of β-lactamases and *mcr-1* genes has not been extensively studied in Ecuador. In this study, we investigated the presence of this combination of resistance genes in two rural communities in the amazon and the coast of Ecuador. Therefore, this study aimed to examined for six types of ESBL and four carbapenemase genes coexisting with the *mcr-1* gene in *E. coli* and *K. pneumoniae* isolates from both humans and their backyard animals in rural low-income communities of Ecuador.

## 2. Results

The majority of colistin-resistant *E. coli* and *K. pneumoniae* isolates (harboring *the mcr-1* gene) in our study were found to co-harbor with eight different β-lactam resistance genes. Specifically, 95% [87.8–100] 95% CI of all isolates (n = 59/62) carried at least one β-lactam resistance gene across the three hosts included in the study. For individual *E. coli* isolates, co-harboring of *mcr-1* and β-lactam resistance genes was observed in 100% of human (n = 10/10) and pig (n = 9/9), and 80% [49–100] 95% CI of the chicken (n = 9/11) isolates. For *K. pneumoniae*, co-harboring of *mcr-1* and β-lactam resistance genes was observed in 100% of chicken (n = 3/3) and pig (n = 6/6) isolates, as well as in 91% [75–99] 95% CI of the human isolates (n = 21/23). Notably, we only detected the carbapenemase gene *bla_KPC_* in one *K. pneumoniae* isolate, which was found in co-existence with another β-lactam resistance gene.

### 2.1. Genomic Coexistence of β-Lactamase and Mcr-1 Resistance Genes

For *E. coli*, 30 *mcr-1* positive colistin-resistant isolates, 7 ß-lactam–resistance genes were detected, with *bla*_TEM_ being the most frequent in 24 positive isolates (80%); followed by *bla*_CTX-M-9_ in 6 isolates (20%), *bla*_CTX-M-1_ in 5 (17%), and *bla*_OXA-48_ in 2 (7%). Finally, *bla*_NDM_ or *bla*_CTX-M-8/25_ was present in only a single isolate. No isolates were positive for *bla*_CTX-M-2_, *bla*_VIM_, or *bla*_KPC_. Table 1 lists the detailed frequency and percentages for *E coli*.

Among the 32 colistin-resistant *K. pneumoniae* isolates that tested positive for *mcr-1*, we detected the coexistence of seven different β-lactam resistance genes. The most prevalent resistance gene was *bla*_SHV_, which was found in 27 isolates with 84% [67.8–100] 95% CI, followed by *bla*_TEM_ in 19 isolates (59%) [37–81.7] 95% CI and *bla*_CTX-M-9_ in 4 isolates (13%) [0–17.4] 95% CI. We also found *bla*_CTX-M-1_ in three isolates (9%), which co-existed with *bla*_CTX-M-8/25_ in one isolate, and with *bla*_NDM_ and *bla*_KPC_ in one isolate each. Notably, none of the isolates were positive for *bla*_CTX-M-2_, *bla*_VIM_, or *bla*_OXA-48_ (Table 2).

Figure 1 and Figure 2 provide a comprehensive overview of the phylogenetic relationships, resistance genes, and hosts of the *E. coli* and *K. pneumoniae* strains isolated from healthy humans, pigs, and chickens. It is noteworthy that *K. pneumoniae* demonstrated a higher prevalence of the coexistence of β-lactam resistance genes than *E. coli*. *K. pneumoniae* LR63 from a healthy human in Santo Domingo de los Tsachilas showed the highest number of resistance genes with four different β-lactam resistance genes. Additionally, 4 isolates had three, 15 isolates had two, and 10 isolates had one β-lactam resistance gene. Intriguingly, a sample (PSJ22) obtained from a healthy human showed the presence of both an *E. coli* and *K. pneumoniae* strain with two β-lactam resistance genes each: *bla*_CTX-M-1_, *bla*_TEM_ in *E. coli*, and *bla*_SHV_, *bla*_TEM_ in *K. pneumoniae*, respectively.

### 2.2. MLST Analysis

From the 7 conserved genes of the 62 isolates (30 from *E. coli* and 32 from *K. pneumoniae*), 434 sequences were obtained (Appendix A). From those sequences and the construction of allelic profiles, 27 different strain types (ST) were identified (15 for *E. coli* and 12 for *K. pneumoniae*), whereas 35 isolates (15 *E. coli* and 20 *K. pneumoniae*) neither presented matches for any ST in the Pasteur MLST database. The minimum-spanning tree (MST) illustrates the classification and distribution of each strain type detected. For *E. coli*, we observed that ST10 was the central one for most strains. Accordingly, that strain type is the node that interconnects a large part of the strains (Figure 3). In addition, two strain types (ST98 and ST10) share the same clonal complex (CC), CC10.

The MST shown in Figure 4 establishes the relationship between the STs of *K. pneumoniae*. In this case, the ST founder is ST504. Two isolates shared the sequence type ST2370. For the majority of the *K. pneumoniae* strains, the ST was not found in the Pasteur MLST database. (See also the limitations of this study.) The phylogenetic analysis showed intermixing between strains of animals and humans. Examples are clones ST400, ST1263, and ST2370.

## 3. Discussion

Our study represents the first report of the coexistence of the *mcr-1* gene with ESBL and carbapenemase loci in two major bacterial pathogens, *E. coli* and *K. pneumoniae*, detected in backyard animals and healthy humans in rural Ecuadorian communities. We found a high frequency of ESBL genes in both bacterial species, with the *bla*_SHV_, *bla*_TEM_, and *bla*_CTX-M_ as the most prevalent genes, and only a single co-occurrence of carbapenemase genes. 

### 3.1. Co-Harboring of Resistance Genes

Our investigation revealed that the distribution of ESBL and carbapenemase resistance genes was similar across hosts, including humans, chickens, and pigs, as well as between two provinces, Pastaza, and Santo Domingo de los Tsachilas, located in the Amazon and Coastal regions of Ecuador, respectively. These results suggest a general dispersion of these genes within nonhospital settings regardless of region or host. Our findings are consistent with several other studies conducted in South America, which have also identified β-lactam resistance genes in various enterobacteria and hosts [23,28,29].

The ESBL genes have been found historically in clinical isolates in Ecuador of Enterobacterales, such as *E. coli* [30], *K. pneumoniae* [31], Acinetobacter baumanii [32], and Raoultella ornithinolytica [33]. Others studies in Ecuador indicated the presence of β-lactam-resistance genes in food animals [27,34] and in the environment [6,18].

The distribution of ESBL genes varies in different bacterial species. In particular, *mcr-1* and *bla*_TEM_ were found to coexist most frequently in *E. coli*, while *bla*_SHV_ was the most common in *K. pneumoniae*. Previous research has also suggested a higher prevalence of TEM enzymes in *E. coli* compared to other ESBL enzymes. Studies in South America have shown that the *bla*_TEM_ variant is more prevalent in turkeys, chickens, and humans in Brazil [35,36] and Ecuador [34], respectively. However, in Germany and Iran, 93% and 51,3% of ESBL-positive *E. coli* strains isolated from adult urinary tract infections were of the *bla*_SHV_ type, respectively [37]. In Libya, the *bla*_TEM_ gene was detected in 30% of the isolates, the *bla*_OXA_ gene in 43% isolates, and the *bla*_CTX-M_ gene in 26% isolates in humans; but, this study analyzed a relatively low number of 23 ESBL-positive *E. coli* strains [38]. 

Our study showed a high incidence of *bla*_TEM_ in *E. coli*, which contrasts with other studies that have reported *bla*_CTX-M_ as the most prevalent ESBL variant [39,40,41,42]. This difference may be attributed to the varying effectiveness of CTX-M enzymes in different epidemiological environments [43,44]. A recent study conducted in broiler farms in Ecuador found that *bla*_CTX-M-9_ and *bla*_CTX-M-1_ were the most common sublineages [27]. However, this was not observed in our study and, instead, *bla*_TEM_ was found to be more prevalent in our samples. It is worth noting that a previous study conducted in Venezuela reported *bla*_CTX-M-1_ as the prevalent ESBL gene in healthy children [45], which is also inconsistent with our findings. This disparity may be due to differences in the origin and types of samples used, with previous studies focusing on clinical settings and urban areas. Additionally, the low frequency of screening at the interface of backyard animals and humans may contribute to differences in the epidemiology of ESBL genes [23]. Our results suggest that resistant clones and/or resistance elements in *E. coli* strains may be geographically segregated.

In *K. pneumoniae*, the most frequently detected ESBL gene combination was *mcr-1* and *bla*_SHV_ (n = 27/32), which is in agreement with previous studies that have found *bla*_SHV_ to be the most dominant β-lactamase gene worldwide, followed by *bla*_TEM_ [46]. In our study, we only found six *bla*_CTX-M_ genes in *K pneumoniae*, highlighting the dependence of ESBL gene prevalence on location, community, and sample type. The lack of reports on ESBL and carbapenemase genes in *K. pneumoniae* in non-hospital settings may bias the estimation of their real prevalence [47].

It is possible that the incidence of *bla*_CTX-M_ in Ecuador has decreased over time due to a lack of selective pressure towards these genes. However, it is important to note that this hypothesis cannot be confirmed without previous studies on the human–backyard–animal interface. Another explanation for the prevalence of *bla*_TEM_ and *bla*_SHV_ in *E. coli* and *K. pneumoniae*, respectively, compared to *bla*_CTX-M_, could be the greater selection and maintenance of these genes. This selection may be due to a plasmid addiction system that contributes to the stability and maintenance of the former genes [48,49]. Plasmid adaptations, such as the presence of a multicopy plasmid linked to *bla*_TEM_ and *bla*_SHV_ genes, are associated to fitness increase under conditions of strong selection for β-lactam resistance and provide an evolutionary advantage by the horizontal gene transfer [50,51].

Within this context, it is noteworthy that the selective pressure exerted by colistin use in livestock on these bacteria does not significantly influence by the selection of other antibiotic resistance groups such as *bla*_CTX-M_ or carbapenemases, including *bla*_OXA-48_, *bla*_KPC_, *bla*_NDM_, or *bla*_VIM_, unlike β-lactams like cefotaxime. This independence could partly explain the low incidence of these types of enzymes [52]. Although colistin use has been banned in Ecuador since 2019, our findings suggest that bacteria circulating within the human–backyard–animal interface reflect the state of antimicrobial resistance in the rural environment and act as reservoirs for the spread of not only *mcr* genes, but also other plasmid-mediated antimicrobial resistance loci such as ESBLs and carbapenemases.

### 3.2. MLST Analysis

The MLST analysis for *E. coli* conducted in this study revealed a high heterogenicity, where ST10 was found to be a central node. This clone has been previously reported in various sources, including humans [53,54], pigs [55], chickens [56,57], sewage [58], and rivers [23]. ST10, belonging to CC10 (clonal complex 10), is globally a highly prevalent clone and has been identified as one of the main reservoirs of *mcr-1*, particularly in fecal and food samples [59]. A 2017 global phylogenetic analysis of *E. coli* found that ST10 was the most common clone identified in 40 out of 312 isolates from different continents [60]. Furthermore, an emerging clone of foodborne extraintestinal pathogenic *E. coli*, ST10, was found to comprise 50% of isolated samples in another study [61]. These findings suggest that ST10 is a widely dispersed founding strain across various environments and hosts.

Furthermore, ST98, also a clone previously described in Ecuador, was found to be closely related to ST10 and belongs to CC10 and shown to be highly transmissible across various hosts, including water sources, food, diseased cattle, pigs, and humans [18]. Other *E. coli* strain types (3856, 5855, 10596, 5123, ST226, and ST5848) identified in this study have been linked to the presence of *mcr-1* in other studies and reported in animal such as pigs, poultry, cattle, and ducks, as well as in healthy humans (farm workers) [57,61,62,63,64,65,66].

Likewise, the diversity of *K. pneumoniae* clones identified in this study reflects their heterogeneity. The founding clone was ST504, reported in a clinical sample of a human [67], and the others are derived from this central strain type. ST2370 was the only strain type with two isolates and has also been reported in clinical fecal samples from outpatients in China [68]. Clones ST4143 and ST1263, identified in this study from animals, have also been reported in hospital areas such as intensive-care units and hematology, and neonatology services in China and France [69,70]. The *K. pneumoniae* clones ST540 and ST3441, detected in our study in human samples, have been identified in studies from patients with pneumonia in China [71] as well as in patients related to a suspected outbreak of bubonic plague in Madagascar [72]. Another *K. pneumoniae* clone, ST4631, which was identified in this study from pig fecal samples, has been previously reported in Sweden in patients with sepsis caused by an ESBL-producing *K. pneumoniae* [73]. Overall, these findings highlight the adaptability and potential dissemination routes of these clones, emphasizing the importance of monitoring antimicrobial resistance in both clinical and non-clinical settings.

## 4. Conclusions

In conclusion, our study has identified a concerning co-occurrence of β-lactam and *mcr-1* resistance genes in Enterobacterales species, posing a significant threat to public health in the communities we investigated. The *bla*_SHV_, *bla*_TEM_, and *bla*_CTX-M_ were the most prevalent genes. Of particular concern is the presence of these strain types in animals raised for human consumption, which could contribute to the transmission of MDR (multidrug-resistant) strain types to humans. As such, there is an urgent need for increased surveillance and epidemiological control to prevent the emergence of antibiotic resistance and implement timely intervention strategies. Low-income rural areas in Latin-American countries, where backyard animal husbandry is prevalent [74], are particularly vulnerable to the spread of antibiotic-resistant pathogens. Of special concern is the presence of several ST identified in this study that have been previously associated with epidemiologically relevant clones in the hospital and non-hospital environments, highlighting the need for further surveillance addressed to control the spread of MDR bacteria and underscores the need for a coordinated One-Health approach to combat the global threat of antibiotic resistance.

Despite the ban of colistin use in animal husbandry in Ecuador since 2019, the continuous use of other antibiotics, such as β-lactams in agricultural production, may contribute to the permanent co-harboring of these genes in animal and human carriers [74]. This, in turn, increases the risk of zoonotic transmission of resistant bacteria from animals to humans, given the close coexistence between humans and backyard animals. Hence, the enhanced surveillance of antibiotic usage and resistance patterns in both human and animal populations is necessary to mitigate this risk. It is essential to raise awareness and implement effective measures to mitigate the spread of antibiotic-resistant pathogens in animal and human populations. Failure to act decisively risks a continued rise in antibiotic resistance, resulting in increased morbidity and mortality, and threatens the foundations of modern medicine.

## 5. Material and Methods

### 5.1. Bacterial Isolates

The colistin-resistant isolates used in this study were obtained from a previously collected bacterial collection at Universidad de las Americas, originating from a study that aimed to investigate the prevalence of antibiotic-resistant bacteria in various hosts within two rural localities of Ecuador: La Reforma and Santo Domingo de los Tsáchilas (coastal region), and San José and Samasunchi, Pastaza (Amazon region) [74]. In total, 30 isolates of *E. coli* and 32 of *K. pneumoniae* that harbored both the *mcr-1* gene and a minimum inhibitory concentration of ≥ 4µg/mL were selected for analysis (Appendix A).

### 5.2. DNA Extraction

The DNA extraction of the isolates was performed by following the Chelex-100 protocol with certain modifications [75]. To obtain a high quality and quantity of DNA, 6–10 colonies of each isolate were resuspended on 200 µL of 10% (*w*/*v*) Chelex (Sigma-Aldrich, USA). After vortexing for 2 min, 10 µL of proteinase K, 20 mg/mL (Invitrogen, Carlsbad, CA, USA) were added. The samples were next centrifuged at 10,000 rpm for 2 min and incubated in a water bath at 56 °C for 1 h. The samples were then vortexed for 5 min and centrifuged in the same manner for 2 min to continue with a second incubation in a thermoblock (Sigma-Aldrich, St. Louis, MO, USA) at 96 °C for 20 min. After another centrifugation at 10,000 rpm for 3 min, the supernatant that contained the DNA was transferred to a clean 1.5 mL microtube and stored at −20 °C.

### 5.3. DNA Quantification

The concentration and purity of the DNA were determined by means of a Thermo Scientific™ NanoDrop™ (Thermo Fisher Scientific, Waltham, MA, USA). Replicate samples of bacterial DNA were placed and a reading performed at 260 and 280 nm to obtain the concentration in ng/µL. In addition, the 260/280 ratio is at the same time a measure of the purity. The sample was then diluted to a concentration of 10 ng/µL.

### 5.4. Molecular-Genetic Identification of β-Lactamase-Encoding Genes

For β-lactamase-encoding genes detection, a single-endpoint polymerase chain reaction (PCR) was performed for six ESBL genes (*bla*_TEM_, *bla*_SHV_, *bla*_CTX-M1_, *bla*_CTX-M2_, *bla*_CTX-M-8/25_, and *bla*_CTX-M-9_) with the primers as described by Le et al. [76]. Furthermore, for detection of the carbapenemase genes *bla*_OXA48_, *bla*_KPC_, *bla*_NDM_, and *bla*_VIM_, a single-endpoint PCR was performed according to previously described procedures [77].

We analyzed the PCR products in a 2% (*w*/*w*) agarose gel using SYBR™ Safe (Invitrogen, Carlsbad, CA, USA) and 1X TBE buffer. The electrophoresis was programmed at 100 V for 35 min in a Labnet Enduro Gel XL horizontal chamber (Labnet International, Inc., Edison, NJ, USA). The agarose gel was visualized in a gel-documentation system (ChemiDoc™ Imaging Systems BioRad, California, USA) and through the use of Image-Lab™ (BioRad, California, USA) software. The length of each amplicon observed in the gel was determined by comparison with a DNA ladder of 100 bp (Invitrogen, Carlsbad, CA, USA). The DNA-positive controls for the various genes were kindly donated by the Osaka Institute of Public Health, Japan. A non–ESBL-producing *E. coli* strain was used as a negative control (*E. coli* ATCC 25922). The PCR products were sequenced, and sequence identities were confirmed by blast analysis.

### 5.5. Multilocus Sequence Typing (MLST)

In order to determine the strain types circulating between the three hosts of the study, the MLST technique was performed according to the indications of the Pasteur MLST database (http://pubmlst.org). The PCR conditions of seven housekeeping genes—*adk*, *fumC*, *gyrB*, *icd*, *mdh*, *purA*, *recA* for *E. coli*, and *gapA*, *tonB*, *rpoB*, *phoE*, *mdh*, *infB*, *pgi* for *K. pneumoniae*—were amplified and sequenced as described by Wirth et al. (2006). The PCR products were sequenced by the Sanger sequencing technique in an ABI 3500xL genetic analyzer (Applied Biosystems, USA) and a BigDye 3.1^®^ capillary electrophoresis matrix. Moreover, the allelic profile and strain-type determination were also confirmed through the use of the above PubMLST website. Finally, the MLST analysis was performed by constructing minimum-spanning trees (MSTs). The construction stated in brief: the PHYLOViz online software (https://online.phyloviz.net/index, accessed on 2 February 2023) was used to analyze sequence-typing methods from allelic profiles and epidemiologic data by means of the goeBURST algorithm, which is based on the determination of descent patterns for bacteria based on the number of differences between allelic profiles [78].

### 5.6. Phylogenetic Analysis

For visualization of the co-harboring resistance genes, an evolutionary history was inferred by using the maximum likelihood method and the Tamura-Nei model for both bacteria species. Evolutionary analyses were conducted in MEGA X (Philadelphia, PA, USA), and the tree was analyzed and annotated using iTOL platform [79]. Initial tree(s) for the heuristic search were obtained automatically by applying Neighbor-Join and BioNJ algorithms to a matrix of pairwise distances estimated using the Tamura-Nei model, and then selecting the topology with superior log likelihood value. Evolutionary analyses were conducted in MEGA X, and the tree was modified using iTOL.

## 6. Limitations of This Study

The present study is subject to certain limitations. Firstly, the ESBL and carbapenemase resistance genes were not characterized phenotypically and, therefore, a relationship between their presence and antimicrobial susceptibility patterns on the phenotypic level was not established. It is worth noting that discordant results have been reported previously for strains harboring genetic resistance elements and their phenotypic susceptibility profiles. Despite detecting the presence of these genes through molecular biology and sequencing, the expression of the enzymes and the levels of resistance in vivo may vary [80].

Furthermore, while the classical seven-genes MLST was used to analyze all strains in the study, no ST could be assigned to most of our strains due to the limitations of this method. Nowadays, whole-genome sequencing (WGS) is the standard technique for bacterial genotyping and has proven high discriminatory power. Therefore, the use of WGS in future studies could provide more accurate and detailed information about strain classification and genetic diversity.

## Figures and Tables

**Figure 1 antibiotics-12-00856-f001:**
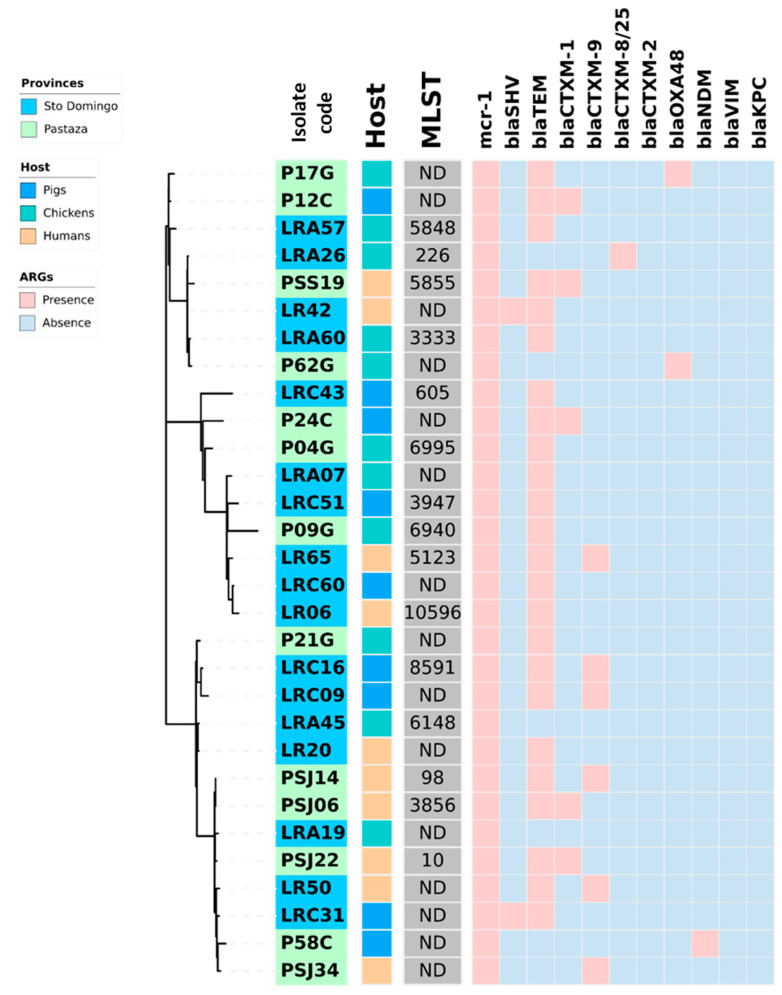
Heat map showing the coexistence of β-lactamases genes and *mcr-1* in 30 *E. coli* strains isolated from fecal swabs from three hosts (healthy humans, pigs, and chickens) of two rural areas (Pastaza and Santo Domingo de los Tsachilas) of Ecuador. In the center of the figure a phylogram of concatenated sequences of 7 housekeeping genes (*adk*, *fumC*, *gyrB*, *icd*, *mdh*, *purA*, *recA*) for the 30 analyzed *E. coli* strains is shown. The evolutionary history was inferred by using the maximum likelihood method and the Tamura-Nei model. The tree with the highest log likelihood (−14986.75) is shown. The lab number of the strains is indicated in the first inner circle with the origin of the strains in blue (Santo Domingo) or green (Pastaza). The host of the strains, pigs, chickens, and humans, is indicated in the second inner circle with, respectively, a blue, green, or orange color. Abbreviations: MLST, multi-locus sequence type in grey; ARG; antibiotic-resistance genes.

**Figure 2 antibiotics-12-00856-f002:**
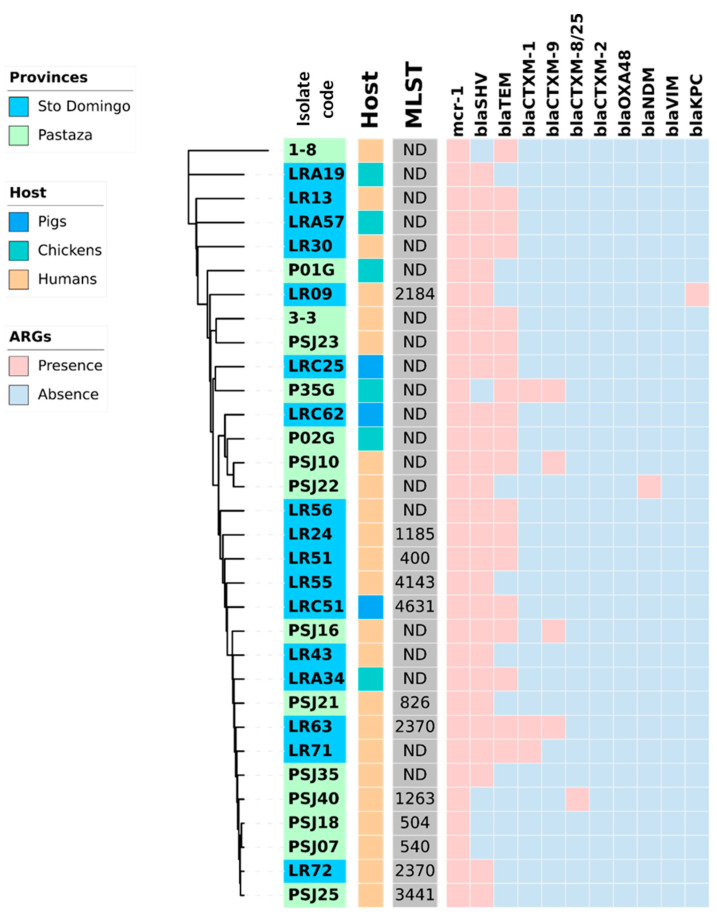
Heat map showing the coexistence of β-lactamases genes with the *mcr-1* gen in 32 *K. pneumoniae* strains isolated from fecal swabs from three hosts (healthy humans, pigs, and chickens) and two rural areas (Pastaza and Santo Domingo de los Tsachilas) of Ecuador. In the center of the figure a phylogram of concatenated sequences of 7 housekeeping genes *(adk*, *fumC*, *gyrB*, *icd*, *mdh*, *purA*, *recA*) for the 32 *K. pneumoniae* strains is shown. The evolutionary history was inferred by using the maximum likelihood method and the Tamura-Nei model. The tree with the highest log likelihood (−14986.75) is shown. The lab number of the strains is indicated in the first inner circle with the origin of the strains in blue (Santo Domingo) or green (Pastaza). The host of the strains, pigs, chickens, and humans, is indicated in the second inner circle with, respectively, a blue, green, or orange color. Abbreviations: MLST, multi-locus sequence type in grey; ARG; antibiotic-resistance genes.

**Figure 3 antibiotics-12-00856-f003:**
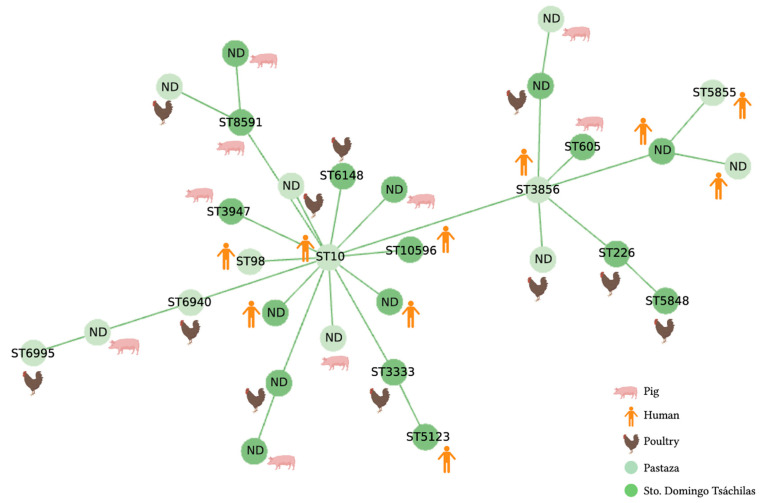
Minimum spanning tree (MST) generated using the PHYLOViz tool, from the allelic profiles of 30 *mcr1*-positive *E. coli* isolates. Allelic profiles were constructed according to the 7-genes scheme (*adk*, *fumC*, *gyrB*, *icd*, *mdh*, *purA*, *recA*) from the PubMLST database. Each ST is represented by a circle. The lengths of the lines between each ST proportionally demonstrates the number of different alleles. Each circle is labeled with the corresponding ST or with the designation ND (not determined). The samples were collected in two provinces of Ecuador, Santo Domingo (dark green circle) and Pastaza (light green circle) between March and December 2019. Based on SNP differences, most strains were derivates of the *Escherichia coli* ST10 and ST3856 clones.

**Figure 4 antibiotics-12-00856-f004:**
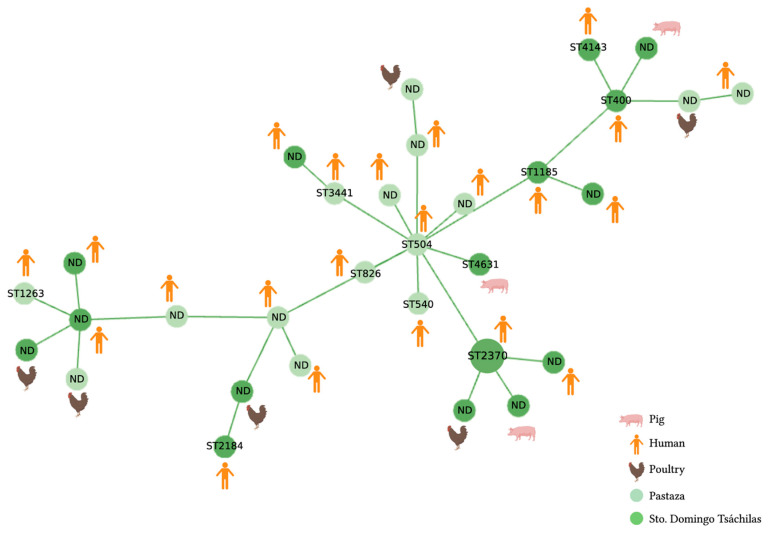
Minimum spanning tree (MST) generated using the PHYLOViz tool from the allelic profiles of 30 *mcr1*-positive *K. pneumoniae* isolates. Allelic profiles were constructed according to the 7-genes scheme (*tonB*, *phoE*, *rpoB*, *pgi*, *infB*, *gapA*, *mdh*) from the Institute Pasteur database. Each ST is represented by a circle. The lengths of the lines between each ST proportionally demonstrate the number of different alleles. Each circle is labeled with the corresponding ST or with the designation ND (not determined). The samples were collected in two provinces of Ecuador, Santo Domingo (dark green) and Pastaza (light green) in March and December 2019. Based on SNP differences, most strains were derivates of the *K. pneumoniae* ST504, ST400, and ST2370 clones.

**Table 1 antibiotics-12-00856-t001:** Frequencies and percentages of *mcr-1*-positive *E. coli* isolates coharboring β-lactamase genes from three different hosts in rural communities of Ecuador (N = 30). * Frequency, (percentage), [95% CI].

Host	*bla* _SHV_		*bla* _TEM_		*bla* _CTXM-1_		*bla* _CTXM-9_		*bla* _CTXM-8/25_		*bla* _OXA48_		*bla* _NDM_	
Healthy humans (n = 10)	1 * (10)	[0–34.4]	9 (90)	[65.5–100]	3 (30)	[0–67]								
Pigs (n = 9)	1 (11.1)	[0–38]	8 (89)	[62.1–100]	2 (22.2)	[0–57]	2 (22.2)	[0–57]					1 (11.1)	[0–38]
Chickens (n = 11)			7 (63.6)	[26.2–100]					1 (9.1)	[0–31.4]	2 (18.1)	[11.8–48]		
Total isolates (N = 30)	2 (6.6)	[0–18.2]	24 (80)	[61–98]	5 (16.6)	[1–34.1]	6 (20)	[1.19–39]	1 (3.3)	[0–11.7]	2 (6.6)	[0–18.2]	1 (3.3)	[0–11.7]

**Table 2 antibiotics-12-00856-t002:** Frequencies and percentages of *mcr-1*-positive *K. Pneumoniae* isolates coharboring β-lactamase genes from three different hosts in rural communities of Ecuador (N = 32). * Frequency, (percentage), [95% CI].

Host	*bla* _SHV_		*bla* _TEM_		*bla* _CTXM-1_		*bla* _CTXM-9_		*bla* _CTXM-8/25_		*bla* _NDM_		*bla* _KPC_	
Healthy humans (n = 23)	19 * (82.6)	[62.2–102]	12 (37.5)	[11.5–63.5]	2 (8.7)	[0–23.8]	3 (13)	[0–17.3]	1 (5.3)	[0–17.3]	1 (5.3)	[0–17.3]	1 (5.3)	[0–17.3]
Pigs (n = 3)	3 (100)	[-]	3 (100)	[-]										
Chickens (n = 6)	5 (83)	[43.5–100]	4 (66.6)	[17–100]	1 (16.6)	[0–55.7]	1 (16.6)	[0–55.7]						
Total of isolates (N = 32)	27 (84.4)	[67.8–100]	19 (59.4)	[37–81.7]	3 (9.4)	[0–22.7]	4 (13)	[0–17.4]	1 (3.1)	[0–17.36]	1 (3.1)	[0–17.36]	1 (3.1)	[0–17.36]

## Data Availability

Not applicable.

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
