# Peer review of "Co-Harboring of Beta-Lactamases and mcr-1 Genes in Escherichia coli and Klebsiella pneumoniae from Healthy Carriers and Backyard Animals in Rural Communities in Ecuador"

_antibiotics, 2023, doi:10.3390/antibiotics12050856_

Round 1

Reviewer 1 Report

The authors have addressed an important global issue: multidrug resistance among bacteria. Every year, thousands of deaths occur to multi-drug resistance strains of bacteria along with burden of billions of dollars to the healthcare system. In this work, the authors have addressed the co-occurrence of extended spectrum β-lactamase (ESBL) and carbapenemases genes in Escherichia coli and Klebsiella pneumoniae strains carrying the mcr-1 gene two rural communities in Ecuador. Interestingly, the others found that 95% of the mcr-1 isolates contained at least one b-lactam gene. The co-occurrence of these two-resistance gene cassette should be kept at watch as these can turn into pandemic by acquiring the horizontally acquired genes.

I’ve tow minor comments:

1.    Figure 1 is difficult to read because of the arc like nature of the heatmap. The authors should modify it to a straight column like structure.

2.    Figure 2 is difficult to read because of the arc like nature of the heatmap. The authors should modify it to a straight column like structure.

Reviewer 2 Report

I read the manuscript entitled "Co-harboring of Beta-lactamases and mcr-1 genes in Escherichia 2 coli and Klebsiella pneumoniae from healthy carriers and 3 backyard animals in rural communities in Ecuador'. In general, the introduction and discussion part should be improved in terms of writing and referencing. A map of the study location is recommended for clear understanding by the reader. Before further processing manuscript needs major revision.

Abstract

The abstract should follow the instructions of the journal. It should be non-formated. Methods in the abstract line number 27, seem the results instead of methods.

Introduction

Line 42 t0 47: Use appropriate references for this part

Line 101 t0 104: Why results are here. Highlights the study objective at the end of the introduction

Results

Table 1: Add 95%CI in the table and the text of lines 106 to 115.

Table 2: Add 95%CI in the table and the text of lines 127 to 133.

Discussion

Most of the parts are in general discussion. Add your key findings and discuss only those key findings. 

Methods

4.1. Delete 163 to 165 as this ethical approval part is repeated in the methods and ethical approval section at the end of the manuscript.

4.2, 4.3, and 4.4: add company names and addresses with locations for all the chemicals and instruments used for the test

Delete table 3 as all the primers have already been published. Use the text and references for more understanding.

4.6: use the address and location same as 4.2, 4.3 and 4.4

Conclusion

Focus only on the key findings and recommendations

Reviewer 3 Report

The authors aimed to determine the coexistence of extended-24 spectrum β-lactamase (ESBL) and carbapenemases genes in Escherichia coli and Klebsiella pneumoniae strains carrying the mcr-1 gene two rural communities in Ecuador. The experimental design, results and conclusions are solid and well-presented. However, there have been several research studies on the co-presence of beta-lactamases and mcr-1 genes in bacteria isolated from animals, such as Lima et al, antibiotics, 2022; Huang et al., Emerging Microbes and Infections, 2020; and Nguyen et al., Frontiers in Veterinary Science, 2022. The authors didn’t provide enough justification on the novelty of their study.

Round 2

Reviewer 2 Report

The manuscript improved well after the revision but still recommends English and plagiarism checking.

Reviewer 3 Report

The study aimed to determine the presence of extended-spectrum β-lactamase (ESBL) and carbapenemases genes in Escherichia coli and Klebsiella pneumoniae strains carrying the mcr-1 gene in rural communities in Ecuador from healthy humans and backyard animals. The authors identified 28 different sequence types, with most never described in humans and animals, indicating that backyard animals may serve as a reservoir of mcr-1/β-lactams resistant genes. These findings are alarming as they threaten the efficacy of last-resort antibiotics. The presentation of figures in the manuscript is well improved. 

Author Response

We appreciate your comment and your interest in our study.